# Effects of Early Nutrition Factors on Baseline Neurodevelopment during the First 6 Months of Life: An EEG Study

**DOI:** 10.3390/nu15061535

**Published:** 2023-03-22

**Authors:** Dylan Gilbreath, Darcy Hagood, Graciela Catalina Alatorre-Cruz, Aline Andres, Heather Downs, Linda J. Larson-Prior

**Affiliations:** 1Arkansas Children’s Nutrition Center (ACNC), Little Rock, AR 72202, USA; 2Department of Neurobiology and Developmental Sciences, University of Arkansas for Medical Sciences (UAMS), Little Rock, AR 72207, USA; 3Department of Pediatrics, University of Arkansas for Medical Sciences (UAMS), Little Rock, AR 72207, USA

**Keywords:** EEG, infant diet, breastfeeding, spectral power, source localization, gamma, resting state

## Abstract

Throughout infancy, the brain undergoes rapid changes in structure and function that are sensitive to environmental influences, such as diet. Breastfed (BF) infants score higher on cognitive tests throughout infancy and into adolescence than formula fed (FF) infants, and these differences in neurocognitive development are reflected in higher concentrations of white and grey matter as measured by MRI. To further explore the effect diet has on cognitive development, electroencephalography (EEG) is used as a direct measure of neuronal activity and to assess specific frequency bands associated with cognitive processes. Task-free baseline EEGs were collected from infants fed with human milk (BF), dairy-based formula (MF), or soy-based formula (SF) at 2, 3, 4, 5, and 6 months of age to explore differences in frequency bands in both sensor and source space. Significant global differences in sensor space were seen in beta and gamma bands between BF and SF groups at ages 2 and 6 months, and these differences were further observed through volumetric modeling in source space. We conclude that BF infants exhibit earlier brain maturation reflected in greater power spectral density in these frequency bands.

## 1. Introduction

Infancy is marked by the rapid emergence of cognitive, behavioral, and social-emotional functions that have been shown to be sensitive to environmental influences during this critical period of development [1]. Infant diet is increasingly recognized as crucial for optimal myelination [2], neurogenesis [3], structural development of early anatomical architecture [4,5], and cognitive development [4,6,7]. As such, the influence of diet on neurodevelopment could have lifelong effects on the structure and function of the brain. While the structural and functional effects of specific nutrient deficiencies, such as iron [4,8,9] and docosahexaenoic acid (DHA) [3,9], on the developing brain have been well-studied in infants, the more subtle differences resulting from infant feeding behaviors are not yet fully elucidated.

Exclusive human milk feeding until 6 months of age and continued human milk feeding for the first two years of life is recommended by the American Academy of Pediatrics (AAP) [10,11] and is widely regarded as being the optimal nutrition source in infants for cognitive development. According to a 2018 CDC report, only 25.8% of infants are exclusively breastfed until 6 months, with many mothers either supplementing or exclusively using formula [11]. Infants who are formula fed (FF) are then recommended dairy-based formula (MF), with soy-based formulas (SF) being the last choice, which is often made because of dietary constraints [10]. While infant formulas contain similar micronutrients and macronutrients inherent in human milk, such as short-, medium-, and long-chain polyunsaturated fatty acids (PUFAS), iron, phospholipids, choline, and DHA, the exact composition and concentration of these essential nutrients can vary [4]. Cognitive differences have been documented between breastfed (BF) and FF children, measured by a slight but significantly higher intelligence quotient (IQ) and Bayley Scales of Infant Developmental (BSID) scores in children who were BF [12,13,14], and these effects were shown to persist into adolescence [7]. Higher IQ in BF children may be confounded by the fact that in high-income countries, breastfed children normally come from a higher socioeconomic group than formula-fed children, and these cognitive differences may reflect a higher parental educational level [12]. However, studies controlling for this potential confound have found similar results with the BF infants scoring higher on cognitive exams later in life [15,16,17]. These findings together suggests that BF infants have a neurodevelopmental advantage, yet cognitive tests alone cannot give insight into the neurophysiological underpinnings responsible for such differences in cognitive outcomes.

Advancements in imaging modalities have made it possible to evaluate brain development in infants. Numerous studies in MRI have demonstrated greater white [4,14] and grey [18,19] matter volumes in infants who were BF compared to MF or SF infants, and these studies suggest that nutritional differences in the commercial formulas are the primary cause. While MRI provides a good basis for the structural effects of diet on the brain, the temporal dynamics and underlying neuronal activity is understudied. The brain is a complex system of rhythmic activity, and these rhythms or oscillations modulate mental experience and determine how the brain processes environmental input [20]. These rhythms develop in infancy and have been used to better understand the emergence of learning [21,22], language processing [23], and the general development [24] of neural networks in infants. EEG is a direct measure of this neuronal activity—representative of the population firing of neuronal ensembles—and is reflective of underlying cognitive processes. While EEG gives excellent temporal resolution on the order of milliseconds, its coarser spatial resolution is a known limiting factor. However, using source space modeling algorithms allows for a more accurate model to determine where the electrical signals originate from the cortex [25] and gives greater insight into the topological patterns of spectral power. In this way, EEG can measure the spectral dynamics of the developing brain, giving greater insight into the maturation of underlying neuronal processes while also giving an increasingly accurate spatial resolution with the implementation of source space modeling. Because this study is interested in spectral dynamics and its implication for neuronal maturation, EEG is the ideal modality.

It is thought that the emergence of oscillatory frequency bands, known as spectral power, mirrors the underlying maturation of cortical networks [26,27]. These frequencies in adults are delta (2–4 Hz), theta (5–7 Hz), alpha (8–12 Hz), beta (15–29 Hz), and gamma (30–45 Hz). In general, spectral power in higher frequencies, such as beta and gamma, increase with age, while spectral power in lower frequencies, such as delta and theta, tend to decrease into early adulthood [28]. Higher frequency band power is associated with cognitive processing [23,27], although the majority of studies examining the power spectrum in infants and children focuses on the lower spectra, with alpha being the most commonly studied frequency as it correlates strongly to known visual cues [29]. The power spectrum is actively developing during infancy, and lower frequencies predominate for the first decade of life; so, some studies shift the canonical adult frequencies to those that have known behavioral and functional correlates in infants [30]. Many studies in infants and early childhood, however, choose to preserve the adult frequencies for two reasons: (1) the literature concerning when and how the power spectrum develops is sparse, and (2) both activation and co-activation of higher frequency bands in the adult range, such as beta and gamma, have been shown in infants [31,32]. This co-activation or cross-frequency-coupling has been demonstrated in infants in the one- to three-month age range, with beta and gamma coupling observed as a mechanism for early speech discrimination [32]. While the power in gamma and beta is low during infancy, it does exist and has been shown to be responsible for a number of developing cognitive processes, including the perceptual binding of objects [31]. For these reasons, this paper will use the adult power spectrum to remain consistent with the literature and to avoid potential changes in these frequencies as we compare them across development.

The primary aim of the current study was to determine whether differences arose in the power spectra in both sensor and source space reconstructions between BF, MF, and SF infants at 2, 3, 4, 5, and 6 months of age using a high-density 128 channel EEG. Because previous work in diet and cognition reports a slight cognitive advantage in BF infants [6,13,17,19,23,33], we predict that the BF group will have a greater global concentration of the high frequency bands associated with cognitive processing—beta and gamma—across age ranges, and this concentration will be reflected in the prefrontal cortex in our source reconstructions.

## 2. Materials and Methods

### 2.1. Participants

Data were collected from 536 healthy term infants (>37 weeks gestational age, between 2.73–4.09 kg) that were enrolled in the Beginnings study (www.clinicaltrials.gov URL last accessed on 21 March 2023, ID#: NCT00616395), a longitudinal cohort study examining the effect of infant diet on physiological and cognitive development. Infants were recruited between 1 and 2 months of age, and as a result many missed their 2-month-old visit for EEG collection leading to fewer participants in this particular age group. Parents selected to exclusively provide their infants a BF, MF, or SF diet. MF and SF standard formulas were supplemented with DHA and arachidonic acid to better mimic the nutritional composition of breastmilk [34]. To qualify, infants must remain on the same diet from the age of 2 months, and mothers of enrolled infants are reported to have abstained from alcohol, tobacco, and/or medications while both pregnant and during lactation. Each infant stayed on the same diet until 12 months of age, with complementary foods optionally introduced after 4 months of age. Other exclusion criteria include voluntary withdrawal at any point during the study, failure to obtain usable EEG data due to excessive artifact, developmental or neurological disorders, and a change in selected diet after the age of 2 months. The total group composition of infants whose EEG data were analyzed is summarized in Table 1. Informed consent was obtained from parents prior to study participation, and the study’s protocol was approved by the Institutional Review Board of the University of Arkansas for Medical Sciences.

### 2.2. Anthropometrics and Behavioral Assessments

Anthropometric measures (height, weight, and head circumference) and infant diet history were obtained during each visit. Gestational age, birth weight, and birth length were also obtained per parental reports. Licensed psychological examiners conducted behavioral assessments on both the infants and their mothers. The infants underwent the second edition of the Bayley Scales of Infant Developmental (BSID) at ages 3 and 6 months to obtain the mental developmental index (MDI) and psychomotor development index (PDI) [35], and the mothers took the second edition of the Wechsler abbreviated scale of intelligence (WASI) to derive their full-scale IQ score during the 3-month visit [36].

### 2.3. EEG Recordings and Signal Processing

Eyes-open high-density (128 channel nets) EEGs were collected in infants at ages 2, 3, 4, 5, and 6 months old. EEGs were collected during a task-free video baseline for each age group over the course of approximately 5 min. Although a silent video played to promote wakefulness, these data are considered resting state EEGs and are in line with the current literature [24]. EEGs were preprocessed in Matlab using the standard Harvard Automated Processing Pipeline for Electroencephalography (HAPPE) [37], in which data were band-pass filtered (0.5–45 Hz), bad channels were rejected, and artifacts were removed via wavelet-enhanced thresholding and ICA with automated component rejection. The HAPPE was designed to clean pediatric EEGs, which are known to be noisier than adult EEGs, and this standardization should improve reproducibility across studies. EEGs were then segmented in 10 s epochs, and segments were rejected if artifacts exceeding ±200 amps remained. Data were then re-referenced to a global mean using the references electrode standardization technique (REST) [38]. EEGs containing >70% bad channels or segments were rejected as were EEGs with an R Pre/Post wavelet thresholding value below 0.2 for our frequency range of interest as calculated by the HAPPE. A minimum of 10 artifact-free segments per subject was required for subsequent analysis using the Brainstorm software package [39].

A standardized infant brain atlas [40] was used to calculate the boundary element head model for each subject age, and sensor locations were projected along the surface generated in line with standard fiducials. Power spectral density (PSD) was calculated using Welch’s method over 1 s epochs with 50% overlap and averaged across all 128 sensors to provide a global metric for the following frequency bands: delta (2–4 Hz), theta (5–7 Hz), alpha (8–12 Hz), beta (15–29 Hz), and gamma (30–45 Hz). The PSDs were then normalized by their relative power in each frequency band. Noise covariance matrixes were calculated for each subject from an individual epoch, and the diagonal noise covariance matrix was used for source estimates that were calculated per subject using the minimum norm method sLORETA [25]. Normalized PSD values were then calculated in source space using the same methodology and frequency bands as sensor space.

### 2.4. Statistical Analyses

The effects of the dietary group on each frequency band, infant BSID scores, and maternal WASI scores were determined by an analysis of variance (ANOVA) for each age group. Post hoc t tests were used for detecting differences between means of the individual dietary groups, and significance was set at *p* < 0.05. To control for potential covariates for the observed effects of the dietary group on PSD, a secondary analysis was preformed using a general linear model to explore the interaction of biological sex with the following between-subjects measures: gestational age, weight at birth, maternal WASI score, and head circumference at the time of the EEG. Multiple comparisons were corrected using Sidak’s method, and significance was set at *p* < 0.05. Statistical testing was accomplished using SPSS Statistics 28.

## 3. Results

### 3.1. Anthropometrics and Behavioral Assessments

A significant main effect of the dietary group was not observed for the BSID, MDI, and PDI assessments at 3 months of age, which is consistent with a previous study’s findings [13]; however, it was observed at 6 months of age for the MDI (*F* = 3.0, *p* = 0.049). Post hoc tests observed significant differences in BF vs. MF (*p* = 0.027) and BF vs. SF (*p* = 0.041) for the MDI scores and a trend toward significance between BF vs. SF (*p* = 0.051) for the PDI score for the six-month age group. Height, weight, and head circumference were also assessed at each EEG visit, with height and head circumference being significant solely at 6 months of age. Infant weight was significant at 2, 3, 5, and 6 months of age. In addition, birth length was not significant in any age group while birth weight was significant at 5 months of age, and gestational age was significant at 3, 4, 5, and 6 months of age. Results are summarized in Table 2. Maternal WASI scores were also calculated at the six-month visit and were found to have a significant effect on infant diet choice (*F* = 11.5, *p* ≤ 0.001) for that age group.

### 3.2. Spectral Power in Sensor Space

Significant differences between the dietary group and delta, theta, and alpha were not observed for any age group. Previous studies using a subset of this data observed regional cortical differences inferred from the EEG sensor placements [33,41]; however, our global analysis exploring gross differences in band power did not observe these effects. At 2 months of age, significant differences in the dietary group were seen in gamma (*F* = 3.215, *p* = 0.04), and further post hoc testing found BF infants had significantly higher gamma than SF (*p* = 0.014). Post hoc testing in the two-month-olds revealed that BF infants also had significantly higher beta than SF (*p* = 0.028). These results are mirrored in the six-month-olds, with BF infants having significantly higher beta (*p* = 0.029) and gamma (*p*= 0.048) than the SF infants (Figure 1). The MF infants did not differ from the BF or the SF infants at any age range or frequency band.

### 3.3. Spectral Power Covariate Analysis in Sensor Space

Adjusting for gestational age, weight at time of birth, head circumference, maternal WASI, and sex revealed significant interactions between sex, dietary group, and PSD. Differences in delta, theta, and alpha were not observed at any ages. Differences in beta were seen in the two-, four-, and six-month-olds, and differences in gamma remain at 2 months of age. In the two-month-olds, female infants had a significant effect of the dietary group on beta (*F* = 3.405, *p* = 0.035) and gamma (*F* = 3.232, *p* = 0.041), and pairwise comparisons revealed this was between BF and SF groups in both cases (beta: *p* = 0.043, gamma: *p* = 0.040). There were also sex differences within dietary groups, with BF females having significantly higher beta (*F* = 5.617, *p* = 0.018) and gamma (*F* = 5.895, *p* = 0.016) than their male counterparts. Analysis of the four-month-olds observed that MF females had significantly higher beta (*F* = 7.541, *p* = 0.006) than MF males; however, there was not a significant effect of the dietary group on PSD at this age. The univariate analysis of the dietary group was shown to have a significant effect on beta at 6 months of age (*F* = 3.288, *p* = 0.038); however, pairwise comparisons did not find significant differences between groups (BF vs. MF, *p* = 0.987; BF vs. SF, *p* = 0.138; MF vs. SF, *p* = 0.051). Significant differences between sex, dietary group, and beta were observed (*F* = 3.069, *p* = 0.48), and these differences were primarily due to MF females having higher beta than SF females (*p* = 0.041). MF females also had higher beta than MF males (*F* = 4.530, *p* = 0.034). There were no significant effects of the dietary group or sex on gamma in the six-month-olds. Importantly, adjusting for covariates decreased our study populations as not every subject had every covariate measure. These results are visualized in Figure 2.

### 3.4. Spectral Power in Source Space

To explore the regional differences in spectral power by the dietary group, a source space analysis was performed for the frequencies in each age range found to be significant at the sensor level. In the two-month-olds, this analysis revealed the greatest concentration of prefrontal activation of both gamma and beta in the BF infants, with the least prefrontal activity in the SF infants (Figure 3A,B). Higher prefrontal activation in these frequency bands is consistent with our results in sensor space and provides a more accurate spatial reconstruction of the regional activation seen in previous studies [41]. Source space reconstructions for the six-month-olds revealed similar patterns of higher prefrontal activation in higher frequencies for the BF groups compared to the MF and particularly the SF groups (Figure 4A,C). In addition to the higher concentration of power in the prefrontal cortex, higher temporal beta in both hemispheres is seen in the six-month BF infants (only right hemisphere shown) compared to the other groups (Figure 4B). Source reconstructions also reveal a slight hemispheric asymmetry, with more beta/gamma power in the right hemisphere in the six-month-olds across dietary groups (Figure 4).

## 4. Discussion

This study is the first to track diet-related changes in the entire resting state global power spectrum over the first six postnatal months of life in both sensor and source space. Although we did find significant changes globally in beta and gamma in the two- and six-month-olds, the lack of significance in other age ranges and frequency bands suggests that nutrition has a specific effect on neurodevelopment in these two critical periods. Furthermore, our covariate analysis observed that these differences may also be largely driven by biological sex. These neurodevelopmental differences seen in the electrophysiology are reflected in both the cognitive and motor assessments as measured by the BSID assessment, with the BF infants exhibiting slightly higher scores on both behavioral assessments. In addition, infants in this study were all healthy, and general developmental results should map onto studies investigating spectral content throughout infancy that did not consider diet as a confound to neural maturation.

### 4.1. Age-Related Development of Higher Frequencies

Many studies acknowledge the importance of the first 1000 days after birth for neurodevelopment, with nutritional factors being seen as the primary factor for optimization [1]. From birth until early adulthood, neurodevelopment consists of two driving factors: progressive, including myelination, neuronal and glial proliferation, and synaptogenesis; and regressive, including apoptosis and synaptic pruning. These factors are often concentrated in temporally distinct periods [42]. The period from birth to 3 months of age sees the greatest amount of volumetric growth [43], which is correlated with an increase in synaptogenesis, neurotrophin serum levels [44], γ-aminobutyric acid (GABA)-ergic neurons [45], myelination [2,4,5,14], and the overall proliferation of both glial and neurons in the brain [46]. Throughout these significant and dynamic changes, the brain is particularly sensitive to nutritional deficits [2]. Early nutrition is known to affect neuroanatomy, neurochemistry, and neurophysiology because of the substrates it provides for the synthesis and activation of growth factors [47]. The primary research focus in studies of neuronal maturation has centered on the structural neuronal architecture that emerges as a function of aging and/or diet, with less emphasis on the resultant functional changes. These functional changes can be evaluated using non-invasive methods, such as EEG, with developmental analysis of the power spectrum used to infer development of different populations of neurons. The differences we observed in our unadjusted model at 2 and 6 months of age between BF and SF infants occurred only in beta and gamma frequency bands, which the literature suggests may reflect development of the GABAergic system and the role it plays in functional neural network architecture [48,49].

Coherent high-frequency oscillations in the gamma range are observed as early as the first 3 to 4 weeks of development [27] and are known to increase from infancy to early adulthood [50]. These high-frequency oscillations are known to facilitate the release of neurotrophins, such as a brain-derived neurotrophic factor (BDNF), which is essential for the survival and proliferation of immature neurons [51,52]. BDNF is regarded as the primary driver of GABAergic development [53,54], and higher levels of BDNF are observed in infants who are BF than those who are FF [55,56]. BDNF regulates the maturation of GABAergic networks through its role in synaptic development, which in turn controls BDNF levels through the post-synaptic release in a positive feedback loop [57]. These networks of interneurons producing GABA mediate gamma activity [49] and are known to migrate rapidly until 6 months of age, with progressively slowing migration patterns until 2 years of age [58]. These immature migratory GABAergic neurons act as excitatory neurons until they reach their target site at which point GABA will act as an inhibitory neurotransmitter similar to its action in adults [59,60,61]. The early patterns of excitatory transmission increase synaptic development and are thought to be responsible for gamma band activity [49], which peaks at approximately 2 months of age after which is declines steeply [61]. In addition, animal studies exploring the maturation of the GABAergic system revealed earlier maturation in females than in males, which could explain the sex differences observed in our secondary analysis at 2 months of age [62]. We propose that the higher global gamma power observed at 2 and 6 months of age between BF and SF infants in sensor space is indicative of an earlier maturation of the emerging GABAergic system and subsequent excitatory/inhibitory balance of the central nervous system as a result of subtle early nutritional differences.

While gamma is analyzed in infancy—though almost exclusively in terms of its role in perceptual binding [29]—beta has yet to be characterized although it is thought to be generated in a manner similar to that of gamma [49]. Increases in beta power correlate positively with increasing age and are thought to be a marker of neuronal maturation [63]. While gamma is posited to be indicative of cognition and attention, less is known about the role of beta in infancy. One study found that increases in beta are associated with the acquisition of motoric skills during infancy, reflective of one of its roles in the adult motor cortex [64]. Higher beta in infancy is associated with increased attention [65], while lower beta is associated with a decrease in cognitive developmental scores [66]. These findings are consistent with our study in which we found that higher beta in BF vs. SF is associated with higher BSID scores at 2 and 6 months of age. Beta power could therefore be indicative of faster maturing motor and/or attentional network.

Our secondary analysis revealed sex related differences between males and females within dietary groups, with BF females having higher beta and gamma at 2 months of age than BF males, and MF females having higher beta than MF males at 4 and 6 months of age. This effect in beta, particularly in the MF groups, has yet to be explored in the literature, and future studies are needed to explore these differences longitudinally. The effects of sex in infancy and early childhood are often mixed [67], and sex is routinely not considered as a biological variable of interest.

### 4.2. Regional Development of Beta/Gamma Defined in Source Space

We hypothesized that differences in spectral power between dietary groups would be visually distinct in source space, with the BF infants having a higher beta/gamma power in frontal regions compared to SF infants based on previous findings in the literature [31,41]. Our results support this hypothesis, although the differences at 2 months are more apparent than those in the six-month age group. This difference may be due to the increased neuronal migration to the anterior cortex that occurs during this early postnatal period [58,68]. Further, not only is there higher power in beta/gamma bands in the BF two-month-old infants, but there is also a higher degree of disbursement throughout the frontal cortex. This is seen to a degree in the six-month-olds as well, particularly in frontal gamma. Our analysis also revealed a greater degree of temporal beta in the BF infants at 6 months of age. Beta is associated with visual attention in this area in adults [69] and infants [70]; we suggest that the increased regional beta observed at 6 months of age is related to the use of a video to engage the infant’s attention during rest state EEG acquisition and may be indicative of a greater degree of visual attention. If that is the case, this may be an important consideration in future studies as many researchers use a video/visual baseline in infancy, and this has a potential to be a confound in connectivity or coherency studies examining the temporal lobes.

### 4.3. Dietary Effects on Neurocognitive Testing

Our results are consistent with an extensive literature reporting that BF infants score higher than FF on cognitive testing during infancy, and that these effects persist throughout childhood and to adolescence [4,6,12,16,17,18,71]. In the literature, there are two primary theories to explain this observation: (1) The majority of these studies occur in high-income countries, and the decision to breastfeed is heavily associated with a higher socio-economic and educational status such that higher cognitive scores may be more indicative of having access to better prenatal care or being raised in a more enriched environment; or (2) the specific nutritional content of human milk—particularly the lipid fraction—is optimal for neurodevelopment. To circumvent the confound of socio-economic status, researchers have examined the cognitive scores of BF vs. FF infants in lower-income countries in which the decision to breastfeed is independent of income and educational status. A series of these studies took place in Brazil, with studies reporting that BF infants still scored higher on cognitive tests than their FF counterparts [16,17], while a study in the Philippines in which breastfeeding is inversely related to socio-economic status revealed that children at 8.5 years of age who were BF as infants scored several points higher in IQ tests [71]. In addition, a study controlling for maternal educational status and IQ maintains BF infants score higher than FF infants on subsequent cognitive assessments [72]. From these studies, a tentative conclusion that maternal socio-economic and educational status does not play the primary role in infant development can be drawn. However, maternal WASI was still used in our covariate analysis because of its potential to have a confounding effect.

Studies focused on the components of human milk that result in its optimization as an early nutritional source have largely focused on its lipid composition, with conflicting results. Many researchers have emphasized the importance of PUFAs, including DHA, because of their known role in promoting synaptogenesis and myelination [12]. Yet, SF infants supplemented with DHA continue to exhibit lower cognitive scores overall [41,73] and reduced language-related neural responses [74], while an extensive review reported no differences in cognitive outcomes in BF infants who were supplemented with additional DHA compared to BF infants who were not supplemented [75]. These results contrast with the finding that infants fed formula containing higher levels of long-chain PUFAs had better neurodevelopmental outcomes [4]. These conflicting results may be due to an inherent variability in the commercial formula used or may be due to other nutritional factors not yet well established. Recent research examining the positive effect of human milk oligosaccharides on the infant gut microbiome and modulation of the immune system has emerged [76] and has shown a positive effect on cognition in animal studies [77]. Because the composition of these oligosaccharides is unique to humans and not found in formula, human milk oligosaccharides may explain cognitive differences in BF vs. FF infants; however, more research in this area is needed [78]. In addition, human milk is known to have a positive effect on the infant microbiome [79], ultimately resulting in fewer allergies [80] and decreasing the risk for certain pathologies, such as necrotizing enterocolitis [81]; however, the effect of this microbiome on cognition and neurodevelopment has yet to be elucidated.

### 4.4. Strengths and Limitations

Strengths of this study include a large sample size across multiple early developmental timepoints for each dietary group and the implementation of source space modeling, which is relatively rare in the pediatric EEG literature despite it increasing the spatial accuracy of EEG. Although this study is longitudinal in nature, it is important to highlight that each age group did not consist of the exact same participants due to either a missed visit or unreadable data for that timepoint. Additionally, while our study did control for infant diet until 4 months of age, data were not collected on the effect of complementary foods integrated at 4 months of age if parents chose to do so, and the accompanying changes in breastmilk or formula feeding if complementary foods were introduced.

## 5. Conclusions

We observed significantly higher global beta and gamma in BF infants at 2 and 6 months of age at the sensor level, these results were then explored in source space in which regional differences in the frontal cortex were found. Higher beta in the frontal and temporal lobes are new findings for this age group, while the higher gamma observed is largely supported by the literature. Our secondary analysis looking at covariates showed that these findings are largely driven by sex differences. Importantly, our study looked at resting state metrics of neurodevelopment; resting-state EEG analysis is well-suited for developmental longitudinal studies and has the potential to unveil the underlying mechanisms of neurodevelopment [82]. Future directions using these data include increasing time-points until 6 years of age as well as connectivity and coherency studies.

## Figures and Tables

**Figure 1 nutrients-15-01535-f001:**
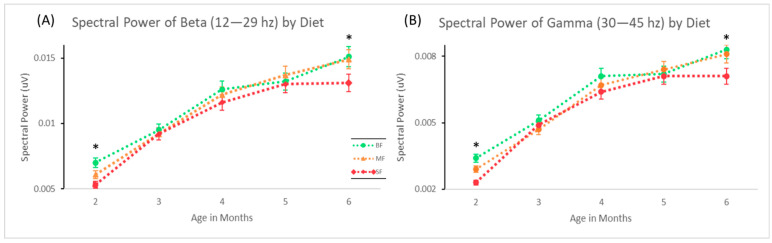
Dietary and age-associated changes in the averaged global spectral power in beta (15–29 Hz, (**A**)) and gamma (30–45 Hz, (**B**)) bands. * indicates a *p* < 0.05 for the BF vs. SF group in both (**A**) and (**B**).

**Figure 2 nutrients-15-01535-f002:**
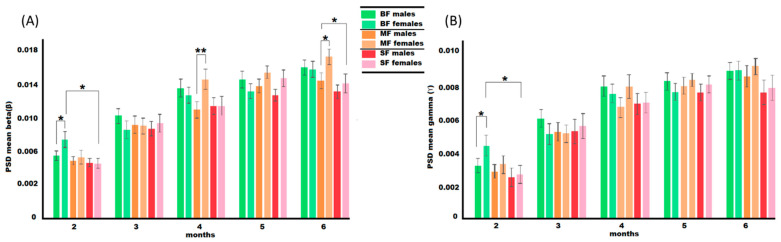
Covariate analysis of the dietary group’s effects on beta (**A**) and gamma (**B**) across 2–6 months of age. Covariates include sex, maternal WASI, gestational age, birth weight, and head circumference at the time of visit. BF: human milk fed; MF: dairy formula fed; SF: soy formula fed. * *p* < 0.05, ** *p* < 0.01.

**Figure 3 nutrients-15-01535-f003:**
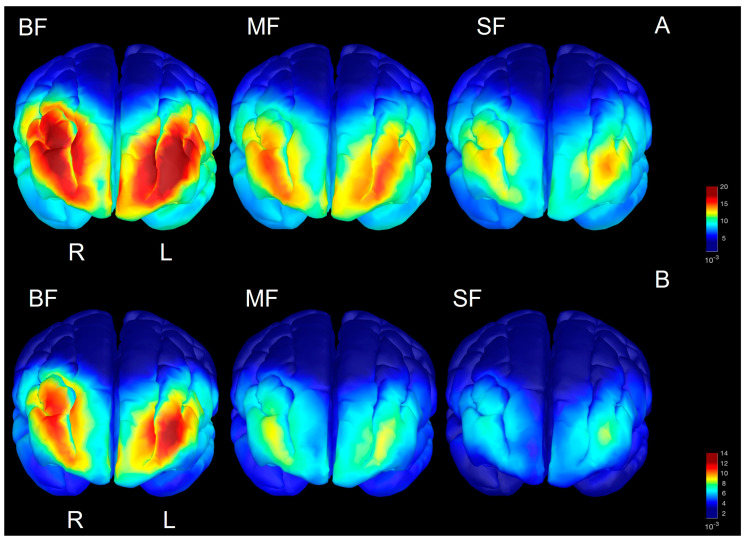
Power spectra distribution of beta (**A**) and gamma (**B**) in two-month-olds based upon dietary group. Greater prefrontal activation is observed in BF vs. SF groups in both high frequency bands.

**Figure 4 nutrients-15-01535-f004:**
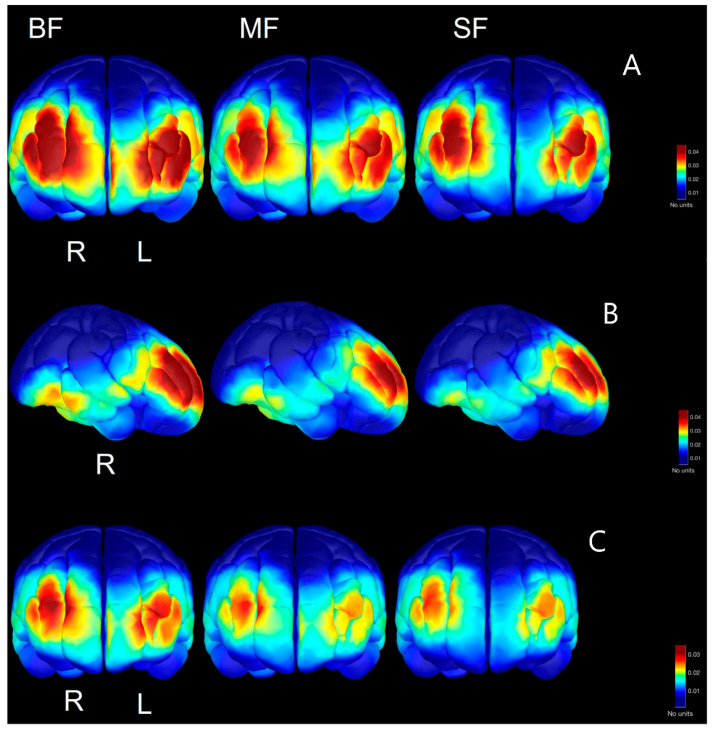
Source space reconstruction of the average power of beta (**A**,**B**) and gamma (**C**) frequency bands in 6-month-olds based upon dietary group. Prefrontal activation is observed in A and C while right, temporal activation is observed in B.

**Table 1 nutrients-15-01535-t001:** Characteristics of Dietary Group by Age.

Age	Total	Dietary Group	Maternal WASI
2 m	313	BF: 108	108.31 (10.76)
		MF: 100	105.54 (9.65)
		SF: 105	102.16 (12.07)
3 m	348	BF: 114	109.34 (10.74)
		MF: 116	105.04 (8.68)
		SF: 119	103.19 (11.49)
4 m	342	BF: 111	109.92 (10.39)
		MF: 110	105.82 (8.90)
		SF: 121	103.66 (10.96)
5 m	417	BF: 137	109.80 (10.05)
		MF: 136	105.16 (9.31)
		SF: 144	103.29 (11.18)
6 m	419	BF: 135	109.78 (10.78)
		MF: 143	105.25 (9.36)
		SF: 141	104.07 (10.96)

m: month; BF: human milk fed; MF: dairy formula fed; SF: soy formula fed. Due to missing data, our secondary covariate analysis had a different number of subjects.

**Table 2 nutrients-15-01535-t002:** Anthropometric Measures by Age and Dietary Group.

Variables	Age	Dietary Group Means (SD)			Main Effect of Group	
		BF	MF	SF	*F*	*p*
Gestation (Weeks)	2 m	39.54 (1.02)	39.19 (0.93)	39.30 (1.03)	4.709	0.1
	3 m	39.65 (1.05)	39.11 (0.96)	39.08 (1.01)	11.626	<0.001 *
	4 m	39.61 (1.05)	39.17 (0.89)	39.31 (1.05)	7.446	<0.001 *
	5 m	39.56 (1.01)	39.14 (0.90)	39.09 (1.06)	8.992	<0.001 *
	6 m	39.60 (1.05)	39.13 (0.90)	39.29 (1.01)	9.807	<0.001 **
Birth weight (kgs)	2 m	3.53 (.032)	3.51 (0.38)	3.42 (0.37)	3.032	0.5
	3 m	3.55 (0.33)	3.50 (0.39)	3.45 (0.36)	2.385	0.094
	4 m	3.57 (0.33)	3.50 (0.36)	3.50 (0.36)	2.914	0.056
	5 m	3.55 (0.34)	3.47 (0.37)	3.43 (0.38)	4.144	0.017 *
	6 m	3.55 (0.34)	3.49 (0.37)	3.45 (0.39)	2.756	0.065
Birth length (cm)	2 m	51.27 (2.14)	51.38 (2.67)	51.11 (2.16)	0.346	0.708
	3 m	51.42 (1.96)	51.08 (2.54)	51.36 (2.10)	0.763	0.467
	4 m	51.58 (2.29)	51.31 (2.50)	51.28 (2.11)	0.572	0.565
	5 m	51.54 (2.18)	51.12 (2.30)	51.22 (2.02)	1.423	0.242
	6 m	51.38 (2.21)	51.24 (2.47)	51.18 (2.14)	0.288	0.75
Height (cm)	2 m	57.39 (1.77)	57.15 (2.07)	56.97 (1.72)	1.351	0.261
	3 m	60.09 (2.01)	60.01 (2.18)	59.68(1.65)	1.708	0.183
	4 m	62.72 (2.01)	62.37 (2.26)	62.68 (2.04)	0.959	0.384
	5 m	64.33 (2.01)	64.52 (2.17)	64.65 (1.92)	0.791	0.454
	6 m	65.86 (2.32)	66.44 (2.38)	66.63 (2.05)	4.289	0.014 *
Weight (kg)	2 m	5.45 (0.57)	5.32 (0.46)	5.17 (0.49)	7.912	<0.001 **
	3 m	6.12 (0.71)	6.13 (0.61)	5.95 (0.52)	3.338	0.037 *
	4 m	6.75 (0.77)	6.77 (0.75)	6.71 (0.64)	0.201	0.818
	5 m	7.18 (0.82)	7.45 (0.86)	7.33 (0.73)	3.967	0.02 *
	6 m	7.63 (0.86)	7.91 (0.83)	7.89 (0.80)	4.835	0.008 **
Head circ. (cm)	2 m	39.38 (1.06)	39.38 (1.08)	39.08 (1.00)	2.73	0.067
	3 m	40.57 (1.11)	40.71 (1.06)	40.60 (1.09)	0.541	0.582
	4 m	41.78 (1.24)	41.77 (1.10)	41.90 (1.14)	0.436	0.647
	5 m	42.63 (1.19)	42.82 (1.10)	42.83 (1.26)	1.187	0.206
	6 m	43.38 (1.27)	43.56 (1.17)	43.78 (1.29)	3.415	0.034 *
	2 m	108.31 (10.76)	105.54 (9.65)	103.16 (12.07)	8.325	<0.001 **
	3 m	109.34 (10.74)	105.04 (8.68)	103.19 (11.49)	10.569	<0.001 **
Maternal WASI	4 m	109.92 (10.39)	105.82 (8.90)	103.66 (10.96)	11.23	<0.001 **
	5 m	109.80 (10.05)	105.16 (9.32)	103.29 (11.18)	14.89	<0.001 **
	6 m	109.78 (10.78)	105.25 (9.36)	104.07 (10.99)	11.5	<0.001 **
		M/F			*x^2*	*p*
Sex	2 m	50/58	49/51	56/49	1.071	0.585
	3 m	55/59	63/53	64/54	3.057	0.548
	4 m	55/56	59/51	67/54	0.821	0.663
	5 m	63/74	71/65	80/64	2.637	0.267
	6 m	62/73	74/69	80/61	3.231	0.199

m: months; BF: human milk fed; MF: dairy milk fed; SF: soy milk fed; Head circ.: head circumference; SD: standard deviation; M: male; F: female; * *p* < 0.05; ** *p* < 0.01.

## Data Availability

These EEG data will be shared on the OpenNeuro database repository.

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
