# Peer review of "Effects of Early Nutrition Factors on Baseline Neurodevelopment during the First 6 Months of Life: An EEG Study"

_nutrients, 2023, doi:10.3390/nu15061535_

Round 1

Reviewer 1 Report

In the current manuscript, the authors examined the impact of human milk (BF), dairy-based formula (MF), or soy-based formula (SF) on temporal EEG dynamics as well as on anthropometric and cognitive outcomes. They observed a specific effect of nutrition on neurodevelopment in 2 and 6-month-olds in terms of beta and gamma as well as cognitive outcomes. This timely paper is of great interest to the readership of Nutrients. I would like to commend the authors for conducting a high quality research and writing up a high-calibre paper. 

I would be happy to see some more content regarding the emerging gut-brain-axis area as well as emotional development (not just cognitive), and the authors opinions regarding whether the gut microbiome can bridge this neurodevelopmental gap reported and if so, how (i.e., future research suggestions). 

Reviewer 2 Report

The Authors performed an observational study, to explore the effect of diet on cognitive development, evaluating the neurological outcome by EEG, used as a direct measure of neuronal activity.

I have some comments before the publication of the manuscript

- Introduction section is too long. Authors examine in depth the EEG in introduction, to my opinion these informations should be added in methods and/or in discussion section. Introduction should be a short paragraph the explain why you decided to perform the study.

- Line 135. What edition of bayley score?

- The results should be revised. The baseline clinical characteristics are missing (GA, birth weight, prenatal and postnatal data, sex, educational level of the parents and all the variables that could influences the neurodevelopment). If there were statistical differences maybe multivariate analysis should be performed for the significant outcome. 

- Table 2. MF vs SF p value? The babies nourished with BF and MF mixed have been excluded? This information is missing or maybe I missed that. In addition, a flow-chart with all included and excluded newborn should be added 

- Conclusion section should be the last paragraph of the article. Strengths (that should be added) and limitations paragraph usually is included at the end of discussion section. Please provide

- Reference format is not the same requested by the Journal (https://www.mdpi.com/journal/nutrients/instructions). In addition, tables should not be added as figure. Please modify

Round 2

Reviewer 2 Report

.